# RETRACTED: CAVVPM: Challenge-Based Authentication and Verification of Vehicle Platooning at Motorway

**DOI:** 10.3390/s22207946

**Published:** 2022-10-18

**Authors:** Muhammad Arslan, Muhammad Faran Majeed, Rana Abu Bakar, Jawad Khan, Shafiq Hussain, Youngmoon Lee, Faheem Khan

**Affiliations:** 1Department of Computing Sciences, University of Sahiwal, Sahiwal 57000, Pakistan; 2Department of Computer Science, Kohsar University Murree, Murree 47150, Pakistan; 3Department of Computer Engineering, Faculty of Engineering, Chulalongkorn University, Bangkok 10330, Thailand; 4Department of Robotics, Hanyang University, Ansan 15588, Korea; 5Department of Computer Engineering, Gachon University, Seongnam-si 13120, Korea

**Keywords:** security, vehicle platoon, access control, challenge–response

## Abstract

As a result of vehicle platooning, advantages including decreased traffic congestion and improved fuel economy are expected. Vehicles in a platoon move in a single line, closely spaced, and at a constant speed. Vehicle-to-vehicle communications and sensor data help keep the platoon formation in place, and the CACC system is responsible for maintaining it. In reality, V2V transmissions are essential for reducing platooning distances while still ensuring their safety and security. It is far more difficult to confirm the veracity of a V2V message’s content than it is to verify its integrity and source authentication. Only platoon members can send and receive V2V communications by implementing a practical access control mechanism. The goal is to link a prospective platoon member’s digital identification to their actual location inside the unit. A physical challenge–response interaction is used in the CAVVPM process to verify that a prospective platoon member respects the rules. The applicant is asked to perform a series of random longitudinal movements, thus, the protocol’s name. Remote attackers cannot join the platoon or send bogus CACC communications because CAVVPM blocks them. CAVVPM is more resistant to pre-recording assaults than previous work, and it can validate that the candidate is precisely behind the verifier in the same lane compared to previous studies.

## 1. Introduction

The term “autonomous platooning” involves the integration of several autonomous vehicles driving in near vicinity across extended distances in a single stream [1,2]. Platooning increases the capacity of the road and fuel economy while preserving safety because of the reduced intervals between vehicles [3,4,5,6,7,8,9]. There is no mechanical connection to maintain the inter-vehicle spacing, which is managed in a synchronized manner. Vehicle-to-vehicle (V2V) communications and onboard sensing chips coordinate steering and acceleration, respectively (cameras, LIDAR, radar). Each platoon member uses a cooperative adaptive cruise control (CACC) algorithm to ensure a safe platoon formation distance and respond appropriately in the face of oncoming traffic. It is significantly easier to identify impending changes in vehicle trajectories with V2V communications than with sensory input. For example, before a change in velocity can be detected, a V2V communication propagates information about a brake application. It reduces the safety space between cars significantly [10,11].

The authenticity of V2V CACC communications is essential to the security of the platoon and the surrounding vehicles. Autonomous platooning technology may be scrapped if a false message is sent into a vehicle’s communication system. A public key infrastructure (PKI) is recommended by wireless standards such as IEEE 1609.2 [12] and 3GPP TS 33.185 for cellular vehicle-to-everything (C-V2X) [13] (PKI). A V2V message’s authenticity and integrity may be verified using cryptographic techniques. The message originator cannot be tied to a specific path.

Remote assaults are possible due to the absence of physical trajectory verification. It is possible even if an opponent claims to be following a platoon from a distance. Communicating with the platoon via C-V2X over the cellular infrastructure is possible. As a legitimate vehicle or a compromised vehicle, the adversary has access to valid cryptographic credentials. Fake communications may be injected into the battalion and disrupt CACC operations once the enemy has authenticated their digital identity. Because the attacker may pose as phantom vehicles in numerous remote areas simultaneously, this assault can be carried out by multiple platoons.

Previous publications have offered physical access control systems to reduce the dangers of remote assaults [14,15,16,17]. Only vehicles that confirm they genuinely follow the platoon should have access to the unit. Proof-of-following (PoF), introduced by Xu et al., was used to formalize the idea in Figure 1. Candidate member C participates in a challenge–response process with a verifier V (usually the platoon’s last vehicle) to link his online signature with his physical trajectory before being accepted into the platoon. A PoF is a complementary approach to digital authentication since it provides physical access control [18,19,20,21]. Since cars currently in the platoon may send bogus messages, this does not prohibit them from doing so.


**Drawbacks of previous approaches**


The convoy methodology [5] correlates the candidate’s and verifier’s trajectories using the vertical acceleration caused by differences in the road surface. However, the static road surface makes the convoy susceptible to pre-recorded assaults. The candidate–verifier trajectories may be correlated using the large-scale fading effect of ambient cellular broadcasts, as described in [22]. To prevent pre-recording assaults, this setting has a high spatial and temporal entropy level. Despite this, the protocol cannot determine the exact distance between the applicant and the verifier; instead, it simply limits the candidate to a specific range around the verifier. Consequently, the candidate might be located within a certain radius of the specified location.

CAVVPM is a novel proof-of-concept protocol that can verify the distance between vehicles of the candidate, the relative placement of the verifier and the candidate, and the candidate’s lane, while also providing resistance to pre-recording assaults. Verification of platoon loyalty is made easy by CAVVPM, which uses a physical challenge–response interaction between candidates and verifiers. The applicant is asked to perform a series of random longitudinal movements, thus, the protocol’s name. A distant adversary cannot react to the verifier’s unexpected motion challenges if it is not within the permitted distance.

**Contributions:** Following is a list of our most significant contributions:PoF protocol CAVVPM relies on physical challenges to secure the system. The purpose of these tests is to link the candidate’s digital identification to his or her path, giving the platoon physical access control. Checkpoints that must be achieved within a certain time frame are part of a physical challenge. In order to verify his path, a foe from afar would have to follow the battalion.Suppose the malicious candidate does not follow the validator within a specified following distance d*_ref_*, not even in the same line as the validator, or is split by any other vehicle. In that case, CAVVPM will not be vulnerable to their assaults. As well as providing lane verification, our protocol establishes a clear order between the candidates and the verifiers. The random nature of the physical tasks makes it even more resistant to pre-recording attacks. It is also immune to man-in-the-middle (MitM) assaults if the verifier’s identity is known to the candidate.Using Plexe’s platooning simulation [12,13,14], we demonstrate that a proof-of-concept verification for critical motorway situations takes some seconds and provides the best security level. In addition, we guarantee that the user feels practically undetectable variations in the vehicle’s velocity throughout the execution of a PoF by employing an ACC algorithm.

## 2. System Model

### 2.1. Platooning Model

We consider a highway platoon of vehicles moving in unison. Vehicles in a platoon are either fully autonomous or semiautonomous, exchanging V2V control messages, including motion state data such as acceleration, velocity, steering, and so on [6]. Many distance-measuring sensors, such as radar, camera, and laser light interferometry (LIDAR), are available in automobiles. A vehicle’s following and preceding distances may be calculated using distance sensors. Platoons use CACC [15,16,17,18] to keep their separation constant based on distance measurements and communications sent.

V2V transmissions are safeguarded using cryptographic primitives to secure the platoon operation. C-V2X communication relies on a public/private key pair (*pk_X_, sk_X_*) and digital certificate (*cert_X_*) provided by a PKI following the standard (3GPP TS 33.185 [1]). Using it, the platoon vehicles may build confidence and validate the source of information. Digital identity and platoon secret key management are outside this project’s scope. It is also explored how non-platooning cars behave in various autonomous platooning circumstances. For this reason, cars that are not platooning are divided into non-priority and priority categories.

### 2.2. Platoon Physical Access Control

To protect mobile platoons from distant intruders, we investigate the issue of physical access control. The primary goal is to limit the number of vehicles in a platoon to those platooning, preventing distant opponents from introducing phony navigation signals. Figure 1 depicts the system model we have developed. An AV platoon is made up of the vehicles AV1 through AV3. To join the platoon, candidate vehicle C claims to be within platooning distance of AV3. The vehicle AV3 verifies C’s trajectory.

**Candidate (C):** Candidate C sends a join request to seek admission to a battalion. An established certificate authority (CA) has granted the candidate a public/private key pair (*pk_C_, sk_C_*) and a certificate (*cert_C_*). It has adaptive cruise control (ACC), which allows him to maintain the following distance on his own without any human assistance. Receiving V2V communications enables the ACC to cooperate. The platoon does not accept a candidate unless it has successfully completed a proof-of-concept challenge from the verifier.

**Verifier (V)**: The platoon’s last vehicle functions as a checkpoint. When a candidate’s digital identity is linked to his or her actual location, it is known as a proof of fact (PoF). A trustworthy certificate authority issues a public/private key pair (*pk_V_, sk_V_*) and certificate (*cert_V_*) to the verifier, just as it does to all other vehicles. In addition, the verifier is capable of accurately determining the distance to any car in the same lane as it.

**The Proof-of-Following Primitive:** Following distance *d_ref_* and the verifier’s lane must be maintained by the candidate. We define PoF in a more rigorous manner than the one first proposed in [22].

**Definition** **1.** **Route:***Time-ordered locations are used to depict a vehicle’s path*
*L_X_*. *L**_𝑋_* = (*ℓ**_𝑋_* (1), *𝑡* (1))*,* (*ℓ**_𝑋_* (2)*,* (*𝑡* (2))*,…,* (*ℓ**_𝑋_* (*𝑛*)*,*
*𝑡* (*𝑛*))*, ℓ**_𝑋_* (*𝑖*) *is the geospatial location of the vehicle*
*(x_X_(i), y_X_(i))*
*with* 1 ≤ *𝑖* ≤ *𝑛**, and*
*𝑡* (*𝑖*) *<*
*𝑡* (*𝑗*) *for*
*𝑖 <*
*𝑗*.

**Definition** **2.** **Proof-of-Following:**
W*e specify a path for the verifier and a way for the candidate. The candidate should follow the verifying driver’s lead in the same lane if Euclidean distance between V and C is satisfied:*

‖ℓVl−ℓCl‖=dreE2∀i
*A desired following distance is defined as*
*d_ref_*
*in this case. If this is not the case, the verifier will REJECT.*

### 2.3. Threat Model

**Adversary goals and capabilities:** *M* is an enemy who tries to go through the PoF verification without following the platoon. The adversary aims to access the platoon and insert bogus cooperation signals. Public and private keys (*pk_M_, sk_M_*) issued by a trustworthy certificate *cert_M_* authority are in the hands of the attacker. Directly or through C-V2X, the enemy may communicate with the platoon.

Figure 2 depicts a distant opponent in cyberspace (a). Opposing forces may be aware of the platoon’s path either before they arrive on the scene or as they go. An imposter car seems to be following them due to his use of the mobile phone network’s architecture. The same situation happens when the enemy is at a distance or ahead of the unit. The adversary does not target the verifier’s range sensors since he is presumed to be far away. The distance sensing technology may be safeguarded even in an assault using a secure ranging protocol [23,24,25].

**Man-in-the-middle adversary.** An attacker can start a man-in-the-middle (MitM) assault while a legal candidate C tries to join the platoon. A MiTM assault is shown in Figure 2b. The opponent prevents the platoon from entering by intercepting C’s platoon entry request and replacing it with his own. *M* pretends to be the verifier to C at the exact moment. Using a PoF, the genuine verifier asks *M* to demonstrate that it is following the instructions. In the PoF protocol, C receives the identical challenge from an opponent.

## 3. The *CAVVPM* Pof Protocol

### 3.1. Overview

Using CAVVPM, the verifier and applicant engage in a physical challenge and response process. The verifier asks the candidate to conduct a set of longitudinal perturbations of its monitoring distance and analyzes these disturbances using the ranging modality to connect the applicant’s digital identity to his physical trajectory. Using the candidate’s public key, the verifier generates a random set of physical challenges and sends them to the candidate. To complete a challenge, they must meet the following requirements: *d_i_*, a desired following distance, and *t_i_*, the deadline.

Candidate C’s responses to two tasks are shown in Figure 3. *d_0_ = d_ref_*, C claims to be in follow-up with V at this exact point in time (to). The verifier determines the following vehicle’s distance and confirms that it is *d_ref_*. A PoF, on the other hand, does not exist if another vehicle follows V. The verifier challenges C to meet deadlines *t_1_* and *t_2_* at checkpoints *d_1_* and *d_2_*. C’s public key is used to encrypt the questions. A “CAVVPM” motion about the following distance *d_ref_* is required for PoF verification. Due to time constraints, it is impossible for a remote attacker to complete the verification process. Relative ordering verification and lane verification may also be accomplished by aiming the range sensor immediately behind the verifier.

### 3.2. The CAVVPM Protocol

The procedure is divided into three stages: digital certificate verification, physical challenge–response, and physical verification. The person’s digital identification is validated in the first step. During the task phase, the verifier asks the candidate to make random movements and then measures the distance between them. The verifier approves the candidate’s claim if he reaches all checkpoints on time during the physical verification phase. As seen in Figure 4, each procedure step is described in detail.


**Digital identity verification phase.**


(1)Candidate *REQ* makes a request for verification V to join the group.

mC1←IDV,IDC,pkC,certC,cigskCREQ, IDC,IDV,


ID_V_ and ID_C_ are the verifier and the candidate’s IDs, (pk_C_, sk_C_) are C’s public/private key pair, and *cert_C_* is C’s certificate.

(2)The verifier uses pk_CA_ to verify the authenticity of certificate C, verifying the signature with pk_C_ the certificate authority’s public key pk_CA_.


**Physical challenge–response phase.**


(3)V creates K physical problems denoted as:
Γ=dref,t0,d1,t1,…,dK,tK,dref,tK+1

a checkpoint *d_i_*, which is a random longitudinal perturbation of the following distance dref, and an associated time limit (*t_i_*) by which the checkpoint must be achieved, are the only two requirements for each challenge The start and end positions of C, which are equal to *d_ref_*, are likewise included in the collection. We use *sk_V_* to sign the challenges before encrypting them with *pk_C_*. As an added bonus, the response’s start time *t_0_* is included in the message.

mC1←IDC,EPKCsigskVΓ,IDV,IDC,to,Γ, IDV,IDC,to


(4)*m_V_* (1) is decrypted by C, and V’s signature is verified by C.(5)After passing through each checkpoint *d_i_* by the deadline *t_i_*, C returns to *d_ref_*.(6)The candidates’ distances are measured and recorded by the verifiers at each deadline. The captured data should be referred to as

Γ′={(d0′,t0),d1′,t1,d2′,t2, … dK′,tK, dK+1′,tK+1}·


**Physical verification phase:** The candidates’ platooning claims are verified in this step by verifying that C has reached the appropriate checkpoints before the deadline.

V analyzes each calculated length 
di′
 to the challenge *d_i_* for each one. In this case, the validator accepts all *d_if_*’s that are inside the acceptable range of 
di′
 to 
γ
. Instead, the validator returns a REJECTION.

∑k=0K+1Idk−dk′≤γK+2=1,


In this case, the indicator function is *I*(.).

### 3.3. Parameter Selection

Checkpoint selection. The verifier calculates a discrete range S around the nominal following distance dref to choose each checkpoint *d_i_*. Assuming UV is the verifier’s velocity, we may write 
gap dref/UV
 in the usual time gap notation to represent following distances of *d_ref_*. We let gmin be the minimum and maximum safety time gaps between any two vehicles. With this, the verifier determines an unbroken range 
gmin·UV,gmax·UV
 for choosing milestone. A discrete range of M checkpoints 
S= s1,s2,…, sM
 is calculated by dividing this range into equal pieces with a length of 2ρ (twice as high as the radar resolution ρ).

M=gmax−gmin· UV2ρ +1


From *S*., a random checkpoint is picked for each task. We consider a verifier moving at *U_V_ = 30* m/s to illustrate the checkpoint selection procedure, as shown in Figure 5 and the simulation setting shown in Figure 6. We let 
gmin=1 s
, 
gmax=2 s 
 with a resolution of radar 
ρ=0.3 m
. Calculations are carried out by the verifier 
M=60m−30m2 · 0.3m+1=51
 checkpoints that are 30 to 60 metres apart. For each candidate, a random checkpoint is selected from among the 51 available by the verifier.

**Deadline selection:** In order to arrive at the given checkpoints safely, the deadlines may be chosen in any way. We consider a relative velocity differential *U_rel_* (positive or negative) to travel the distance 
di−dref
 as an easy approach to pick a deadline for checkpoint 
di
. In this scenario, the time limit is set to 
ti=di−drefvrel +ϵ
, where the candidate’s motion may have a modest amount of tolerance (
ϵ
) built in.

In contrast to the automatic nature of platooning and customer engagement, this simplistic model assumes that the candidate’s velocity changes quickly rather than gradually. Alternately, the verifier may use an ACC model that considers aspects such as safety and motion smoothing to compute deadlines for the system. An optimal control-based intersection CACC (iCACC) system was proposed by Zohdy et al. According to the simulation results, the suggested solution reduces energy consumption by 45% and the average intersection latency by 90%, respectively [12]. A convex optimization problem with linear constraints was later created for a group of coordination leaders and their coordination followers by Van de Hoef et al. to optimize the fuel savings of a Truck CACC system’s coordination leaders [13]. To reduce energy use and pollutant emissions at various phases of the CACC operation, Wang et al. suggested a platoon-wide Eco-CACC system [14]. The intra-platoon vehicle sequence was the subject of another investigation using optimization methods [15]. To penalize relative position mistakes between nearby CACC vehicles, Jovanovic et al. constructed centralized linear quadratic optimal control formulations [16]. They concluded that stability and detectability would decline as the size of the CACC system increased. To determine the ideal engine speed to reduce fuel consumption, Turri et al. explored the cooperative look-ahead control of a heavy-duty CACC system [17].

It is possible to use any ACC controller here; however, we prefer the one provided in [18]. Our ACC parameter in simulation is presented in Figure 6.

The deadline is computed as follows using this model. We allow for a challenge d that corresponds to a time gap of 
T=d/x˙C
, where 
x˙C
 represents the person’s current pace. 
Δt
 is the number of iterations in the algorithm.

(1)At the *n*th step, the required acceleration is

(1)
x¨desn=−1TΔx˙n+λδn


(2)
δn=−dactn+d,Δx˙n=x˙Cn−x˙Vn

where 
Δx˙n
 is the comparison between *C* and *V* in terms of velocity, 
dactn
 is real swaying distance, 𝛿[𝑛] represents the range uncertainty to *d*, and λ > 0 regulates converging to *d*.(2)Speed is applied using input from the preceding phase, rather than 
x¨des n
.

(3)
x¨n=β⋅x¨desn+1−β⋅x¨n−1,β=Δtτ+Δt·
The time interval between the (*n*−1)-st and *n*-th steps is denoted by 
Δt
, where 
τ
 is commonly set to 0.5 s.(3)C’s distance gain during Δ*t* is calculated using this formula.

(4)
ln=x˙n−1⋅Δt+12⋅x¨n⋅Δt2
(4)Step n’s distance [𝑛] to checkpoint d is adjusted to this new value.

(5)
δn=δn−1+ln−x˙Vn⋅Δt
(5)Steps 1 to 4 are repeated until the desired result is achieved 
δn<γ
, In this example, 
γ
 is the checkpoint tolerance. Checkpoint d’s *t* deadline is extended to *t* = Δ*t* ∗ n*. This means that n* is the initial value of n, and |[𝑛]| < 
γ
.

## 4. Security Analysis

### 4.1. Correctness

Following the verification process at *d_ref_*, an applicant who is found to be eligible will obtain from *V* the range of difficulties Γ. Using the ACC to feed Γ allows C to meet the timeframes for the milestones while 
V
 measures his position, finishing the proof of concept. Both the lane number and its relative position are confirmed by 
V
 since its radar modality measures straight rearward inside its own lane.

### 4.2. Remote Adversary

First, we check to see whether an enemy who is not platooning can make it through the proof-of-concept and join the group. The opponent may be anywhere except behind the verifier (A follower of the platoon at *d_ref_* should pass physical access permissions). It is possible that the enemy may be everywhere, including parked far away, driving many vehicles behind the verifier, or even sharing a lane with him. It is possible that the adversary’s car is not the only one following the verifier, but there are two additional alternatives.

#### 4.2.1. No Vehicles Follow Verifier

Adversary *M* may submit a request message m_M_ (1) to the verifier asking to join the platoon as a new member. The defendant will be able to pass the digital identity verification since his certificate is believed to be authentic. To test the adversary’s mettle, the verifier will issue a series of challenges Γ after confirming the adversary’s identity. The validator would be ineffectual to identify a vehicle at the scheduled milestones since the adversary does not follow 
V
 and no other cars follow 
V
.

#### 4.2.2. A Vehicle Follow Verifier

We take into account the scenario in which a vehicle *R* other than the adversary follows the verifier. A safe distance *d_ref_* between the adversary’s opponent and the verifier’s vehicle *R* is maintained, but it is not under the control of the adversary. A requesting message m_M_ (1) is sent to the validator by the malicious user asking to join the platoon. The enemy will be able to pass the electronic identity verification, as previously stated. With help from a series of demands Γ, the verifier will confront the opponent. Instead of the distant opponent, the verifier will estimate the distance to the following vehicle *R* at the deadlines stated in Γ. In order for the PoF to be passed, the opponent must be at the milestones by the appropriate dates set by *R.*

**Using a random walk to model R’s path:** R’s random walk around *d_ref_* serves as the basis for calculating R’s chance of passing the PoF test. While still following, R may move independently of the platoon and change its distance from it within a certain range. Under a certain range [*d_min_, d_max_*], the vehicle R varies its range from the verifier.

N-state Markov chains are used to simulate R’s random walk where states reflect the possible locations of R and transition probabilities indicate the likelihood of moving to a new position within the range after a time step of n. As a result, we limit the range [*d_min_, d_max_*], In this case, we assume that R can traverse a specified distance *d_step_* during a defined time step, and divide the range by *d_step_* to obtain a total of N places (states). The initialization allocation P^(0)^ at time 0 is considered to be regular without losing clarity. The odds of changing states are also provided by a 
N×N
 matrix 
P=Pij
 with:
(6)
P1,1=P1,2=PN,N=PN,N−1=1/2


(7)
Pi,i+1=Pi,i−1=Pi,i=1/3,i=2…N−1


(8)
Pi,j=0, all other i,j·


Figure 6 depicts the randomized walk’s potential range (a). There will always be a transit to a new state in a random stroll, and we have decided to allow the vehicle to remain in the same condition within such a time interval. Furthermore, given a state, the probability of moving ahead, backward, or staying in the same shape is equal, regardless of the value of matrix P. The M potential checkpoints specified by the verifier might not even correspond with the N candidate states of R. However, we may suppose that R’s state space includes M checkpoints. To use the random walk model, we can now calculate the likelihood of clearing the PoF validation, disregarding the beginning and end positions of *d_ref_*.

**Proposition** **1.***Let the verifier put M through a series of K tasks*

Γ=d1,t1,d2,t2,…dK,tK

*In a state space S of size M, each milestone is randomly chosen. A nonlinear model using S ⊆ S′ may be used to find a state space S′ of size N and a vehicle R that follows the verifier. Due to R’s movement, the chance that M passes the PoF verification is provided by:*
(9)
PM=1NMK∏k=1K∑i=1M∑j=1NPj,i∑l=1knk⋅


Following *n* rounds, the transition probability matrix *P^n^* is shown in the following figure.

**Evidence: Passing a single challenge:** Checkpoints and deadlines may be combined into one challenge (d, t). If R is at a distance d at time t, the opponent has won the challenge. Probabilities (X1, X2, etc.) are used to represent R’s non-linear model and the nth simulation time of the discrete stochastic process (DSP).

(10)
Xn=X0⋅P0n


By Pr[pass], we may indicate the likelihood that the opponent will arrive *d* in time step *n*, or, put another way, that the task will be successfully completed. Then:
(11)
Pr[pass]=∑i=1MPr[d=i]⋅PrXn=i∣d=i=1M∑i=1M⋅PrXn=i∣d=i=1M∑i=1M∑j=1NPrX0=j⋅PrXn=i∣X0=j=1NM∑i=1M∑j=1NPj,in


All M potential checkpoint values were taken into consideration in Equation (11). Checkpoints for each task are picked randomly in Equation (11). All of the first N locations for vehicle R were examined in Equation (11), respectively, in the random process.

**Passing a PoF:** A PoF challenger must pass all K challenges in set 
Γ=d1,t1,d2,t2,…dK,tK
 in order to succeed. To put it another way, R must walk to each checkpoint by the given time. With random and independent selection of each checkpoint *d_k_*, the passing rate may be represented as follows:
(12)
PM=Pr[ pass ⋅chal⋅1]…⋅Pr[ Pass ⋅ chal ⋅K]=1NM∑i=1M∑j=1NPj,in1…1NM∑i=1M∑j=1NPj,in1+n2+⋯+nk=1NMK∏k=1K∑i=1M∑j=1NPj,i∑l=1knk


Equation (12) is derived from the difficulties of being independent from one another. Here, we added up the time it takes to complete each task, rather than using the chance of passing a single challenge from Equation (11). For a more condensed form of Equation (12), use Equation (12).

**Lemma** **Proof.**The upper bound on the passage probability of the opponent is

(13)
PM≤1MK


**Proof.** For the purpose of demonstrating Lemma 1, we look at the chance of completing a single challenge, which is given by Equation (11). The summarizing phase should be noted as 
∑i=1MPj,in
. In the *j*-th row of matrix *P^n^*, the *j*th row sums *M* out of *N* elements. Because checkpoints are a subdomain of possible positions for vehicle *R*,

∑i=1MPj,in≤∑i=1NPj,in=1
Equivalently, substituting this limit into Equation (11) results in:
pr[pass]=1NM∑i=1M∑j=1NPj,in≤1NM∑j=1N1=1M
The proof is completed by substituting this bound into Equation (12). □

With respect to the number of checkpoints, we can see that the passing probability PM diminishes exponentially with increasing checkpoint complexity, as shown by Lemma 1. It is possible to manipulate these two factors in order to achieve any desired passing probability, but this comes at a cost in the time it takes for the PoF verification process to finish. When M << N, the limit of Lemma 1 is fairly lenient, as seen in Figure 1. In reality, it is possible to demonstrate that the random walk’s steady-state distribution uniformizes with time (with the exception of the two boundaries that have different transition probabilities). The chance of passing a single challenge is pr[pass] = 1*/N*, which is a variable of M that is under uniform distribution in most of the N states. To put it another way, the likelihood of reaching the chosen checkpoint is one in N. PoF verification has a chance of passing of 
PM∼(1/N)K
 when *K* separate challenges are taken into consideration.

### 4.3. A MiTM Adversary

Attackers are able to access the platoon by impersonating a genuine applicant who has requested to enter the platoon. We look at two examples of the assault to see how it works. To enter a platoon, the applicant uses his public key *pk_V_* and his certificate *cert_V_* to identify a known validator. This applicant tries to disingenuously join an adjacent platoon, but does not specifically seek out the validator.

**Assimilation into a predetermined platoon:** The candidate should be allowed to target the platoon with (ID_*V*_, pk_*V*_, cert_*V*_) from the verifier’s list. In Figure 7A, we see the stages of a MiTM assault. By sending a join request message m_C_(1) to V, the candidate sets up the protocols. Request m_C_(1) includes the person’s private key-signed IDC and IDV. For example, the attacker may try to start parallel connections by blocking m_C_(1) and inserting his personal demand for *V* to join.

mM1←IDV,IDM,pkM,certM,sigskMREQ,IDM,IDV·


The verifier authenticates the digital identity of *M* and challenges *M* with Γ after obtaining *m_M_*(1). Because the opponent is not following the platoon, the only way to effectively finish the MiTM assault is for the legitimate candidate to successfully complete the physical requirements. Γ. In response to C’s first communication mC(1), the adversary may try to transmit by:
mM′1←IDC,EpkCsigskMΓ,IDM,IDC,t0,Γ,IDM,IDC,t0

consisting of the identical collection of physical obstacles Γ that *V* and M put in front of *M.* Because the response is verified by *M* and not *V, C* will terminate the process of joining. The adversary’s MiTM attack fails at this point because C only accepts requests verified by *V*.

**Choosing to join a random platoon:** The applicant might not even know the identity of the verifier when he seeks to join a battalion on the spur of the moment. We consider a candidate who follows V at *d_ref_* but does not know V’s identity. The attacker may then use a MiTM attack in which he spoofs V’s credentials. Figure 7 depicts the processes involved in a MitM attack (b). The applicant C sends mail requesting to join a battalion.

mC1←REQ,IDC,pkC,certC


Keep in mind that the join is not aimed towards any particular verifier (Instead, the candidate may reply to a probe from neighboring verifiers, similar to the receiving of SSIDs from surrounding Wi-Fi networks; however, the final outcome is approximately the same in terms of understanding the name of *V*). *M* corrupts *C* (1) (e.g., by jamming) in order to prevent *V* from receiving it and then opens his own session with *V* by sending him a message.

mM1←IDV,IDM,pkM,certM,sigskMREQ,IDM,IDV


It is the verification. *M* receives a response from the verifier, who says

mV1←IDM,EpkMsigskVΓ,IDV,IDM,t0,Γ,IDV,IDM,t0


This puts Γ on the spot with *M*. As an answer to *m_c_*(1), the opponent pretends to be a validator and challenges the legitimate candidate with the identical obstacles that Γ used.

mM′1←IDC,EpkCsigskMΓ,IDM,IDC,t0,Γ,IDM,IDC,t0


Even if *M* does not follow *V* within the distance *d_ref_*, the candidate performs the perturbations in Γ that induce *V* to accept *M*.

**Resistance to MiTM attacks:** Once the identification of the validator is unknown to the legitimate candidate, avoiding a MiTM attack is difficult. Impersonation is feasible without a way to verify the intended verifier. At this point, we want to underline the technical skill necessary to carry out such a sophisticated assault. The opponent intercepts the request from a legitimate applicant who attempts to enter a platoon advantageously without knowing that a verifier represents the platoon’s identification.

There is no way to avoid this attack using the CAVVPM protocol, but there are a few viable solutions [26,27]. Using highly-focused antennas on both the applicant and the verifier might be a solution (e.g., at mmWave frequencies) by directing the candidate’s transmitter forward when the verifier’s antenna is behind; an attacker has a short window of time to carry out the MiTM. Using a single-receiver transmission localization device to locate the transmitter’s position is another option worth considering (e.g., [27]). It is accurate but used in static situations rather than high-speed mobility environments. The Doppler move between the applicant and the validator might be used to identify messages supplied by the attacker as an alternate approach. This assault can only be used in spontaneous platooning settings, leaving these avenues open for further research [28].

## 5. Evaluation

Here, the CAVVPM system’s safety and efficiency are examined. For overall platooning tests, the Plexe simulation environment [13] was used, which is a cooperative driving framework that allows for realistic modeling of platooning systems. Car interactions and various cruise control models make it possible to analyze traffic in multiple conditions.

### 5.1. Performance of CAVVPM

Before evaluating CAVVPM’s behavior as a result of multiple protocols, we first examined its functionality. We ran a simulation where we accompanied a verifier V and a candidate C on a motorway. The following range was controlled using the ACC model provided in Section 3.3. Table 1 contains a list of the simulation’s parameters. Initially, platooning took place at 30 m/s (108 km/h) with no other vehicles. Distance 
dref
 45 m, which equates to a 1.5 s time delay, was fixed. The verifier sent random physical challenges to test the candidate’s ability to conduct changes at random throughout the preceding distance of 1000 m in 33 s.

**Time for a verification**: Evaluation criteria included the time it took for CAVVPM to accept new recruits. The ACC characteristics and the range of physical difficulties can be used to estimate the amount of time it takes to complete the task.

**Investigating the ACC’s effects:** The ACC settings control threshold deadlines. The vehicle’s speed is controlled by parameter 𝛾, in particular, concerning the checkpoint’s range. At a radius of 3 m from the dref, the checkpoint in Figure 8 indicates the person’s acceleration, speed, and the following field with time. Figure 8A shows that the vehicle’s starting speed progressively decreases until the milestone arrives when it stops. A virtually undetectable change in rate is demonstrated by the speed difference of 0.6 m/s (2 Km/h). In addition, we see a drop in speed and momentum difference with a longer delay until the checkpoint is reached once 𝛾 is reduced to 0.1. We used 𝛾 = 0.4 for the remainder of our runs. Figure 9 displays the Markov chain model, the verifier’s chosen checkpoints, and the N potential states for the vehicle R.

The checkpoint’s range allowance 𝛾 is another crucial factor that affects the latency. When the vehicle is near a checkpoint, it swiftly converges and adjusts its location to arrive at the checkpoint on time. The timeframe may be reduced by raising the distance tolerance. Figure 10A illustrates that 𝛾 is approximately equal to the deadline duration. The 𝛾 = 0.3 m we used in our simulation is close to the standard car sensor resolution [29].

Figure 10A This chart shows how long it will take to complete a checkpoint 3 m distant from dref as a function of the range threshold 𝛾 and (B) the limit time as a function of milestone range, while 𝛾 = 0.3 m.

Figure 10B shows the limit as a function of the distance to the checkpoint. We see that deadlines expand with distance, but the connection is not straightforward. The ACC theory justifies this. We also observe that limits are not symmetric when the same distance must be reached forward and backward due to differing speeds.

**Impact of traffic:** Physical obstacles are seen as a continual source of movement for our verifier. The verifier’s speed and the candidate’s ACC approach to a checkpoint may be affected by traffic. We simulated a car driving at 27 m/s approaching the verifier to examine this effect. While the candidate is trying to reach a checkpoint, V slows its speed to 27 m/s to maintain a safe distance. Figure 11 depicts the candidate’s time-dependent acceleration, velocity, and following distance for a checkpoint 42 m from V. According to Figure 10, the time it takes to reach the checkpoint has risen from 7.6 s to 13.6 s. As a result of V’s braking, the candidate could advance beyond the checkpoint before regaining the checkpoint. In other words, if the verifier’s speed changes, a legitimate candidate will miss the initial deadline.

To solve this issue, there are two ways to proceed. First, if the verifier’s velocity dramatically fluctuates, it is best to disregard any challenges, then repeat them when the velocity stabilizes. Another option is for the verifier to change the deadline depending on his or her own performance. An ACC model may help verifiers calculate the candidate’s checkpoint arrival time.

Time taken for verification in relation to physical difficulties *K*. Depending on how many physical challenges are presented by the verifier, verification time might vary significantly. Because the verification latency increases with each challenge, this connection should be linear. Variations occur because deadlines for randomly picked checkpoints are unpredictable. We experimented with *K* by setting the checkpoint space to *M* = 51 and running CAVVPM with various *K* values. Verification times and their standard deviations are shown in Figure 11A. Verification times have increased linearly as projected, with an overhead of around 10 s each physical challenge. Because the applicant will be platooning with the rest of the platoon for less than a minute, the verification period is rather brief. There are five checkpoints available, and the average verification time for each is shown in Figure 11B. The verification time rises as the candidate’s range of motion widens because the average distance to each checkpoint increases if and only if *K* equals 5. The method validation rises as the person’s motion widens because of the significantly larger range to every endpoint. Figure 12 depicts the candidate’s time-dependent acceleration, velocity, and following distance for a checkpoint 42 m from *V*.

### 5.2. Security of CAVVPM

A virtual opponent cannot pass the PoF verification without fulfilling the challenging situations, as shown in Section 4.2. The only possibility for the opponent is if an independent vehicle *R* follows the verifier at the platooning distance *R*. As stated in Proposition 1, we calculated the likelihood that M will pass verification owing to *R*’s movement. Vehicle R, driving at 30 m/s, was simulated approaching the verifier at the same time. The vehicle *R* performed a random walk with a step size of 0.3 m inside the checkpoint range (30 m–60 m away from the verifier). The verifier provided physical challenges with a tolerance of 0.3 m regularly. *R*’s distance from V as a function of time for five checkpoints is shown in Figure 13A. At the time of the limit, as seen in Figure 13B, *R*’s range from every security check was apparent. We find that *R* often resides far from the designated checkpoint since it makes no effort to go there.

In Figure 14A, *M*’s pass percentage as a result of the number of physical challenges, computed using the formula over 2000 challenges, is shown in greater detail. Note that *M* failed all of the PoFs after *K* = 2, (*PM* = 0). As a point of reference, we have included *PM* as a numerical value derived from Proposition 1. The odds of overcoming even a few minor physical obstacles are quite low. *PM* is unaffected by the checkpoint space attribute *M*. This is because *R* must meet a deadline for a certain checkpoint. The uniform distribution of this probability can easily be demonstrated using a random walk (with slightly higher probabilities for the two boundaries).

## 6. Related Works

**Verification of Platooning.** Many publications have addressed platoon access control [5,7,19]. The convoy is a physical context-based platoon entrance mechanism [5]. The convoy used the link between candidate and verifier vertical acceleration owing to road surface differences. This method is susceptible to record-and-replay assaults since road conditions vary slowly. It cannot determine the distance either. The driving trajectory was employed to prove platoon membership by Vaas et al. [19] and Juuti et al. [7]. A contender recorded its own trajectory as platooning evidence. The platoon trajectory may be known a priori or tracked remotely.

**Physical challenge–response protocols.** A physical task is utilized for security. Shoukry et al. presented PyCRA, a sensor authenticating system [15]. PyCRA injects randomized but recognized physical probe signals to check sensor performance and avoid analog injection attacks. PyCRA can test sensors such as radar range estimators, but it needs radar availability and may not be appropriate for safety-critical applications.

Dutta et al. [4] optimized PyCRA for distance sensors by decreasing the measured-versus-actual distance inaccuracy. Their technique needs to know the distance beforehand, which is unrealistic. Kapoor et al. [8] used spatio-temporal correlations of MIMO antenna broadcasts to solve past system constraints, establishing a spatio-temporal physiological review system [8]. The car’s radar does not need to be switched off during accuracy checks. These initiatives are orthogonal to ours since proving a vehicle’s distance is inadequate to connect it to its digital identity. However, they secure CAVVPM’s sensory mode for verifying physical obstacles. The Table 2 shows related work comparative summary of existing security techniques in platooning system. 

## 7. Simulation Results and Real-World Limitations

An open-source framework based on OMNet++ and SUMO, Vehicles in Network Simulation (VEINS), simulates networks of automobiles. Road traffic is modeled using SUMO, while OMNet++ simulates networks with the help of the physical layer modeling program MiXiM. The multi-channel operation, interference, and noise effects are all accounted for in VEINS’ models of IEEE 802.11p, IEEE 1609.4 DSRC/WAVE network layers, and cellular networks such as LTE and 5G.

Given how cheap it is to alter the network’s topology and protocol, a computer simulation can be used to test out a wide variety of network topologies for vehicle platoons. As a bonus, the effectiveness of the defense mechanism and the platoon’s reaction to an attack may be tested in various traffic scenarios. The vehicle platoon built on the upgraded protocol version can be tested for compatibility with older systems and validated after publishing the new version. While there are many advantages to employing computer simulation, the traffic and network in the model vehicles are inaccurate representations of the real world. Because of this discrepancy between the model and the real world, the predicted behavior of a vehicle platoon may be incorrect, leading to unintended consequences in practice. Simulations platooning results and parameters are displayed in Figure 14 and Figure 15.

## 8. Conclusions

CAVVPM is a physical challenge–response procedure for platoon accessibility. CAVVPM binds a person’s digital identity to his claimed trajectory through arbitrary distance disturbances. CAVVPM may check the candidate’s distance, relative location, lane, and resilience to pre-recorded assaults. We investigated CAVVPM’s security and performance in the Plexe simulator and found that a PoF verification lasts less than a minute while causing nearly undetectable speed changes.

## Figures and Tables

**Figure 1 sensors-22-07946-f001:** A platoon of three vehicles formed by 𝐴𝑉_1_, 𝐴𝑉_2_, and 𝐴𝑉_3_. Vehicle 𝐴𝑉_3_ acts as a verifier V for the candidate C who wishes to be admitted to the platoon. Parties C and V engage in a proof-of-following protocol.

**Figure 2 sensors-22-07946-f002:** The threat model. (**a**) a remote adversary; (**b**) an MiTM adversary.

**Figure 3 sensors-22-07946-f003:** In order to meet the deadlines *t_1_* and *t_2_* set by the verifier, candidates are required to visit randomly generated checkpoints *d_1_* and *d_2_*.

**Figure 4 sensors-22-07946-f004:** The CAVVPM protocol.

**Figure 5 sensors-22-07946-f005:** Setting a range for candidate C’s checkpoint.

**Figure 6 sensors-22-07946-f006:** ACC Parameters.

**Figure 7 sensors-22-07946-f007:** (**A**) ACC controllers Analysis. (**B**) Sinusoidal acceleration analysis.

**Figure 8 sensors-22-07946-f008:** (**A**) Acceleration; (**B**) Velocity; (**C**) Following distance.

**Figure 9 sensors-22-07946-f009:** (**A**) Vehicle *R*’s Markov chain random walk model while (**B**) The verifier selects *M* checkpoints and *R* has *N* potential states.

**Figure 10 sensors-22-07946-f010:** An assault by the MiTM. (**A**) C is familiar with *ID_V_* and *pk_V_*, (**B**) C does not know who the verifier is.

**Figure 11 sensors-22-07946-f011:** (**A**) Checkpoint distance threshold 𝛾 (**B**) the deadline duration when 𝛾 = 0.3m.

**Figure 12 sensors-22-07946-f012:** Shows the change in C’s acceleration, velocity, and distance from *d_ref_* = 45 m to milestone D = 42 m when the verifier’s speed drops from 30 m/s to 27 m/s. (**A**) Acceleration; (**B**) Velocity; (**C**) Following distance.

**Figure 13 sensors-22-07946-f013:** For example, how long it takes to verify a certain number of challenges and milestones. (**A**) Five challenges are used to calculate the distance between R and V. (**B**) Vehicle R’s range from checkpoints at each deadline. (**B**) shows that, independent of *M*, *PM* and (1*/N*)*^K^* are essentially identical, despite the fact that P1 is smaller than (1*/N*)*^K^*.

**Figure 14 sensors-22-07946-f014:** (**A**) time(s); (**B**) Index of checkpoint. Simulated and numerical passing probability *P_M_* vs. challenges *K*.

**Figure 15 sensors-22-07946-f015:** (**A**) Simulation; (**B**) Numerical.

**Table 1 sensors-22-07946-t001:** Simulation Parameters.

Parameters	Values
Velocity of *V* and *C*	30 m/s
*d_ref_*	1.6…𝑣𝐶 (46 m)
Checkpoint range	1…𝑣𝐶 − 2…𝑣𝐶 (31 m–61 m)
# Of checkpoints in range (𝑀)	52
Update step of ACC (Δ𝑡)	0.11 s
ACC parameter λ	0.41
Checkpoint error tolerance (𝛾)	0.31 m

**Table 2 sensors-22-07946-t002:** A Comparative Summary of Security Techniques.

Work	Methodology	Attack	Strength	Weakness
[4]	Cryptography	Eavesdroppingsybil	Lightweight hash-based cryptography	Additional hardware
[8]	Cryptography	EavesdroppingSybil	Elliptic Curve cryptography	Additional hardware needed
[15]	Cryptography	Sybil, Jamming	session key-based certificate	Additional hardware
[19]	Cryptography	JammingImpersonationMessage altering	Certificate Revocation protocols	Additional hardware

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
