# Peer review of "CAVVPM: Challenge-Based Authentication and Verification of Vehicle Platooning at Motorway"

_sensors, 2022, doi:10.3390/s22207946_

Round 1

Reviewer 1 Report

Dear Authors,

The idea of submitted manuscript is up-to-date Paper is well structured, good written and understandable.

Developed model considered only autonomous and semi-autonomous vehicle. Explain how non-autonomous vehicle can be considered on order to implement in model. There is no need for change model, only to explain. This is real situation still. It will improve manuscript.

On the page 12, how are simulation parameters are adopted should be explained. Adopted velocity is 31m/s and verification "lasts less than minute" with respect to safety is length of aprox. 1000m.

Results of simulation should be compared with other published data.

Check typing, for example, on page 6 line 228, subscript are not written properly.

Author Response

Original Manuscript ID: sensors-1868929

Original Article Title: “CAVVPM: Challenge-based Authentication and Verification of Vehicle Platooning for Motorway”

To: Sensors Editor

Re: Response to reviewers

Dear Editor,

I am highly thankful to all reviewers and Editor. Their helpful comments help us to improve the quality of the paper.

We are uploading (a) our point-by-point response to the comments (below) (response to reviewers), (b) an updated manuscript with track changes and clean updated manuscript without highlights (PDF main document).

Best regards,

Muhammad Arslan

Reviewer 1

The idea of the submitted manuscript is up-to-date Paper is well structured, good written and understandable.

Response: Thank you for your response and appreciation. We have done moderate English changes from our institutional English editing services before submitting.

The developed model considered only autonomous and semi-autonomous vehicle. Explain how non-autonomous vehicle can be considered on order to implement in model. There is no need for change model, only to explain. This is real situation still. It will improve manuscript.

Response: Thank you for your response. We have explained how non-autonomous vehicles handle this model by categorising them into non-priority and priority vehicles.

On the page 12, how are simulation parameters are adopted should be explained. Adopted velocity is 31m/s and verification "lasts less than minute" with respect to safety is length of approx. 1000m.

Response: Thank you for your response. We have explained in section 5.1 and added more details on how the velocity of 108KM was achieved.

Results of simulation should be compared with other published data.

Response: Thank you for your response. We have added another section 7 Simulation results.

Check typing, for example, on page 6 line 228, subscript are not written properly

Response: Thank you for your response. We have reviewed typing errors and correct all these issues.

Reviewer 2 Report

Platooning is efficient and powerful for the current traffic congestion decreases and carbon emission reduction. This paper proposes a Wiggle process for prospective platoon members to link their digital identification to actual locations inside a platoon formation. The proposed Wiggle process does a series of random longitudinal movements. The simulation shows the efficiency of the proposed Wiggle method in a Plexe simulation environment.

1. In this manuscript, the proposed Wiggle protocol is possible to use any ACC controller which is claimed by the authors. A widely-used ACC controller [12] is investigated, which is good. If there are some preliminary analyses based on other ACC controllers, there would be more solid for the proposed Wiggle method.

2. For the simulation setup, the speeds of the V and C are around 108 km/h (~31m/s) which is more focused on a typical highway speed scenario. Is there possible to investigate the lower speed cases, like the ramp metering scenario?

3. Please update the presentation regarding some figures and tables according to the author's guide. For example, the fig.9 and 13 can be further edited for better resolution.

4. Last, since the proposed Wiggle method is better than other methods, there would be better if more comparisons are presented between the proposed method and other typical methods.

Author Response

Original Manuscript ID: sensors-1868929

Original Article Title: “CAVVPM: Challenge-based Authentication and Verification of Vehicle Platooning for Motorway”

To: Sensors Editor

Re: Response to reviewers

Dear Editor,

I am highly thankful to all reviewers and Editor. Their helpful comments help us to improve the quality of the paper.

We are uploading (a) our point-by-point response to the comments (below) (response to reviewers), (b) an updated manuscript with track changes and clean updated manuscript without highlights (PDF main document).

Best regards,

Muhammad Arslan

Reviewer 2

Platooning is efficient and powerful for the current traffic congestion decreases and carbon emission reduction. This paper proposes a Wiggle process for prospective platoon members to link their digital identification to actual locations inside a platoon formation. The proposed Wiggle process does a series of random longitudinal movements. The simulation shows the efficiency of the proposed Wiggle method in a Plexe simulation environment.

Response: Thank you for your response and appreciation.

  1. In this manuscript, the proposed Wiggle protocol is possible to use any ACC controller which is claimed by the authors. A widely-used ACC controller [12] is investigated, which is good. If there are some preliminary analyses based on other ACC controllers, there would be more solid for the proposed Wiggle method.

Response: Thank you for your response and we have added preliminary analyses in section 3.3

  1. For the simulation setup, the speeds of the V and C are around 108 km/h (~31m/s) which is more focused on a typical highway speed scenario. Is there possible to investigate the lower speed cases, like the ramp metering scenario?

Response: Thank you for your response and we have designed this protocol for only motorway scenario where average speed 108 km/h. In future we will consider low speed GT roads. Ramp metering will be considered to regulate the upstream traffic flows on highway on-ramps and low speed roads.

  1. Please update the presentation regarding some figures and tables according to the author's guide. For example, the fig.9 and 13 can be further edited for better resolution.

Response: Thank you for your response and we have reformed the Figure 9 and Figure 13

  1. Last, since the proposed Wiggle method is better than other methods, there would be better if more comparisons are presented between the proposed method and other typical methods.

Response: Thank you for your response we have add comparison table in future direction compare our model with existing models.

Round 2

Reviewer 2 Report

Please further respond to the concerns directly in next version. The current version still should be improved according to the initial comments. Only concern three about the figure resolutions was responded with positive revision. Welcome more major revisions of how to resolve these concerns.

Author Response

Original Manuscript ID: sensors-1868929

Original Article Title: “CAVVPM: Challenge-based Authentication and Verification of Vehicle Platooning for Motorway”

To: Sensors Editor

Re: Response to reviewers

Dear Editor,

I am highly thankful to all reviewers and Editor. Their helpful comments help us to improve the quality of the paper. We are uploading (a) our point-by-point response to the comments (below) (response to reviewers), (b) an updated manuscript with track changes and clean updated manuscript without highlights (PDF main document).

Best regards,

Muhammad Arslan
